# A Visual Measurement Method for Deep Holes in Composite Material Aerospace Components

**DOI:** 10.3390/s24123786

**Published:** 2024-06-11

**Authors:** Fantong Meng, Jiankun Yang, Guolin Yang, Haibo Lu, Zhigang Dong, Renke Kang, Dongming Guo, Yan Qin

**Affiliations:** 1State Key Laboratory of High-Performance Precision Manufacturing, Dalian University of Technology, Dalian 116024, China; 825093006@mail.dlut.edu.cn (F.M.); ygl@dlut.edu.cn (G.Y.);; 2Pengcheng Laboratory, Shenzhen 518055, China

**Keywords:** composite materials, image processing, Laplace operator, robotic assembly

## Abstract

The visual measurement of deep holes in composite material workpieces constitutes a critical step in the robotic assembly of aerospace components. The positioning accuracy of assembly holes significantly impacts the assembly quality of components. However, the complex texture of the composite material surface and mutual interference between the imaging of the inlet and outlet edges of deep holes significantly challenge hole detection. A visual measurement method for deep holes in composite materials based on the radial penalty Laplacian operator is proposed to address the issues by suppressing visual noise and enhancing the features of hole edges. Coupled with a novel inflection-point-removal algorithm, this approach enables the accurate detection of holes with a diameter of 10 mm and a depth of 50 mm in composite material components, achieving a measurement precision of 0.03 mm.

## 1. Introduction

Composite materials are widely used in aerospace manufacturing due to their excellent properties, comprising up to 50% of the materials on Boeing 787 aircraft [1,2]. In the assembly process of aerospace components, many composite material parts or components need to be connected under the design requirements [3,4]. Bolting is a major form of assembly connection in aerospace manufacturing [5,6], which requires the preprocessing of assembly holes on the parts or the components to be connected [7]. A total of 5000 to 15,000 assembly holes are typically machined on each aircraft skin panel [8]. Considering the large number of connection holes in aerospace components, automatic drilling is necessary to improve the machining efficiency and ensure assembly quality. Given its advantages, such as the high flexibility of processing, the high consistency of quality, and high normal accuracy, robotic drilling has been applied in the assembly of aerospace components, such as cabin skins and wing panels [7,8,9,10]. Despite the rapid development of automated robotic holemaking, the current bolt-assembly process mainly relies on workers using semi-automatic tools. To improve the quality of automation bolting and overcome the space limitations of manual operation in narrow cavities or tail trusses, robotic bolting technologies and equipment requirements are urgently needed.

In the robotic bolting of aerospace components, inconsistency between workpieces and nominal numerical models is a constant problem due to the workpiece’s processing tolerance, location errors, or structural deformation during clamping, which makes it difficult for robots to fix bolts or nuts in the correct position. Therefore, it is necessary to use visual sensors on robot end effectors to identify and locate the holes during robotic bolting, guide the robots’ movement, and ensure the assembly quality and efficiency of aerospace components. Visual measurement, as a non-contact measurement method with the advantages of high measurement accuracy and convenient integration, is widely used in the on-site detection of workpiece holes [11,12].

Monocular vision, binocular vision, and depth vision are commonly employed in the visual measurement of holes [13,14,15]. Among these, the monocular vision method, with its smaller sensor size, is the most suitable for manufacturing robots, facilitating the in situ measurement of holes. The Hough Transform algorithm is a classical image processing technique widely applied for the detection and localization of hole edges [16,17,18]. Nevertheless, recent progress has prompted enhancements of the Hough Circle Transform and other advanced algorithms aimed at bolstering robustness and improving real-time performance in high-precision hole measurement applications. Yao et al. and Zhao et al. improved the Hough Circle Transform algorithm by leveraging the curvature and gradient distribution patterns of local edges, enhancing computational efficiency, and reducing memory usage [19,20]. Zhao et al. also addressed the requirement for hole detection on metallic components by employing a sub-pixel edge-extraction method based on Zernike moments, thus achieving more precise information on the hole edges [20]. However, the Hough Transform and its improved algorithms still struggle with high computational loads and reduced precision when dealing with noise. In cases where complete and clear edges can be obtained, there are also more efficient hole recognition methods. Chen et al. and Zhang et al. proposed different methods that both rapidly identify candidate circles based on a few edge points, followed by further iterative verification to ascertain the credibility of these candidate circles, achieving a more efficient detection performance compared to that of the Hough Circle Transform [21,22]. Hrechuk et al. proposed a hole contour recognition algorithm based on the Delaunay triangulation method to measure the hole defects of composite material workpieces [23]. In the context of detecting small holes in industrial components, Jiang et al. partitioned the image into blocks, and during the assessment of candidate circles, restricted the edge point selection of a few local image segments, thereby enhancing the detection efficiency [24]. According to the symmetry characteristic of circular hole contours, Zhu et al. proposed a fast hole detection method using the Euclidean distance transformation of a binary image, aimed at determining the precise localization of a large number of holes [25]. Mei et al. proposed a hole measurement method with high accuracy, combining a vote-based ellipse-fitting algorithm with an optimization approach utilizing the snakes model [26]. Nonetheless, the aforementioned methods still failed to overcome the major challenge in the visual measurement of deep holes in composite material workpieces, namely, the problem of incomplete or merged edges due to severe noise interference.

Cavity structures are commonly used in aerospace components, usually found in aircraft cabins, radomes, and aeroengine nacelles. Cavity-shaped components are generally composed of a skeleton and skin, and there are often deep assembly holes at the critical parts with high strength. In some aerospace components, the assembly holes under measurement exhibit a diameter of 10 mm and a depth of 50 mm, resulting in a depth-to-diameter ratio of five. During the robotic bolting of cavity components, fasteners, including bolts and nuts, are grasped by robots and inserted into the interior of the cavity, as shown in Figure 1. To ensure the alignment between the fastener and the assembly hole, visual measurement is performed to locate the assembly hole on the inner wall of the workpiece and guide the robot’s movement. The positioning accuracy of fasteners significantly impacts the joint assembly quality, necessitating a visual positioning accuracy of 0.05 mm for the assembly holes. Within a narrow and dim environment, the visual measurement of assembly holes inside the composite material cavity component faces many challenges. Using an autonomous machine vision framework, Hernandez et al. detected drilled holes on carbon fiber reinforced composite (CFRP) panels over a large range and segmented the delamination features from the holes [27]. Tan et al., using a blacklight-assisted visual measurement system, accurately separated hole edges from curved surfaces in 3D measurement utilizing discontinuity features [28]. Davim et al. separated the hole edge from machining damage features through parameter adjustment and noise suppression when holes were detected on the composite material workpiece [29]. Cui et al. proposed a visual image enhancement algorithm based on threshold segmentation according to the characteristics of visual imaging in order to detect the delamination of composite material assembly holes [30]. A deep learning model was also applied in the visual recognition of holes on composite materials workpieces, typically for the purpose of coarse image segmentation and the extraction of regions of interest (ROIs) [31], and when the deep learning model was meticulously trained with the annotated dataset, the accuracy of hole measurement on the composite material surface could be enhanced [32]. However, in the visual measurement of deep holes in the composite material workpiece, the fine and random texture on the surface of the composite material, which extends to the hole wall, is barely suppressed by the conventional filtering method, causing significant interference in the extraction of the hole edge. Meanwhile, external environment light illuminating through the hole center causes the inlet and outlet edges of the hole to appear simultaneously, with a possible occlusion or connection of their profiles, making it challenging to separate them accurately. In practical applications, current methods still falter when confronting these challenges.

This study proposes a visual measurement method for deep holes based on the radial penalty Laplacian (RPL) operator for the visual detection and location of assembly holes on the inner wall of composite material cavity components. In Section 2, the imaging characteristics of deep holes in composite material components are analyzed, and a second derivative analysis algorithm based on the RPL operator is proposed for images. A systematic measurement method for the inlet edge of assembly holes is presented in Section 3. Section 4 presents the experimental results, and Section 5 presents the conclusion.

## 2. Edge Enhancement Algorithm for Deep Hole Images of Composite Components based on Image Radial Distribution

### 2.1. Imaging Characteristics of Deep Holes in Composite Material Components

In the visual measurement of the assembly hole, the key step is to accurately recognize the target hole edge. Limited by physical space in a narrow cavity, usually a camera with a fixed focal lens and small ring light is used for assembly hole detection, whose imaging follows the pinhole perspective model. As a result, in addition to the target of the inlet edge of the assembly hole, the outlet edge also appears on the image, creating a distraction, as shown in Figure 2.

In aerospace manufacturing, to improve processing performance and reduce burrs and delamination during the drilling process, a rough layer of glass fiber is usually pasted on the composite material workpiece’s surface. The texture of the glass fiber and carbon fiber introduces complex noise, disturbing the detection of the hole inlet edge. A typical deep-hole image of a composite component is shown in Figure 3. According to the pixel distribution, three typical regions can be delivered, including the workpiece surface region with texture distribution, the hole inlet edge region, and the imaging region of the hole outlet edge. The visual measurement of assembly holes is aimed at detecting the hole inlet boundary and accurately fitting it into a circular contour. Inside the cavity component, visual measurement of the assembly hole is affected by environment light illuminated through the hole center, forming a bright area near the image center. Due to the non-overlapping image of the inlet and outlet edge of the hole, a clear edge is caused by the outlet edge. The edge of the hole outlet interferes with visual measurement, which requires a specialized method to separate the two image edges.

The different regions of the assembly hole on the composite component have different pixel grayscale distribution characteristics, as shown in Figure 4. As shown in Figure 4a, in the workpiece surface region with texture distribution, the distributions of pixels along all directions are highly random and lack a clear pattern. As for the hole inlet edge region in Figure 4b, along the radial direction of the assembly hole, the pixel grayscale values change from a severely random distribution to a uniform one. At the same time, the mutation point is the edge of the hole inlet. In the imaging region of the hole outlet edge in Figure 4c, the pixels are uniformly distributed both at the hole wall and the hole center. Only when observed along the radial direction of the assembly hole is there a mutation of the pixel distribution, the mutation point of which is the edge of the hole outlet. Therefore, these three regions can be distinguished according to their different distribution characteristics.

### 2.2. Hole Edge Enhancement Algorithm Based on Radial Penalty Laplacian Operator

The visual detection and measurement of deep holes in composite material components aim to detect hole inlet edge point sets and fit them into a circle or an ellipse. As illustrated in Figure 3 and Figure 4, both the patterns on the composite material surface and the halo at the hole outlet cause disturbances in identifying the hole inlet edge, thereby requiring noise reduction to enhance the hole inlet feature. As a vital step in image processing, the denoising method should be selected based on the characteristics of the image and the specific pattern of noise distribution. The discrete salt-and-pepper noise can be removed through median filtering, whereas periodic noise can be suppressed using Wiener filtering. In more complex global noise scenarios, the BM3D (block-matching and 3D filtering) denoising algorithm [33] represents an advanced strategy. By capitalizing on the similarities in global features among image blocks, BM3D executes 3D block grouping of these blocks and, subsequently, applies denoising filters, thereby effectively combating the interference of global random noise. Built upon BM3D, the CID (Combined Image Denoising in Spatial and Frequency Domains) method incorporates frequency domain similarity of image blocks to enhance noise reduction accuracy even more significantly [34]. The Block-matching Convolutional Neural Network (BMCNN), built upon the block-matching of images, utilizes a CNN model for denoising inference on 3D image block groups, enhancing both the efficiency and accuracy of image denoising [35]. By considering spatial and frequency domain similarity across the entire image, the aforementioned advanced denoising approaches demonstrate remarkable efficacy in suppressing image noise caused by factors such as lens blur or motion distortion. However, a challenge for these algorithms is identifying the intricate patterns on the composite material surface or the smooth edge around the hole outlet halo as noises. There is also a lack of accurately labeled image datasets for deep holes in composite materials, which are crucial for the pre-training of deep learning models. A thorough analysis of the characteristics in images featuring deep holes of composite materials is still imperative, with a particular emphasis on denoising and edge-enhancement methodologies.

Comparing the typical regions of the assembly hole image, the change rate of the pixel grayscale values along the radial direction of the assembly hole is a key feature. When observing the radial grayscale distribution of the image near the hole inlet edge, there is a difference in the change rate of grayscale values on both sides of the hole inlet edge, as shown in Figure 5. The grayscale values range from smoothness to drastic fluctuation, whose dividing line is the hole inlet edge. The crucial step to extract the hole inlet edge is to suppress the grayscale fluctuation other than the hole inlet edge and enhance the features of the hole inlet edge.

The grayscale distribution of the image radially around the hole outlet edge can be seen in Figure 6. The grayscale values change smoothly and decrease suddenly at the hole outlet edge. The sudden variation in grayscale values at the hole outlet edge is similar to the hole inlet edge, which disturbs the hole edge recognition. According to the grayscale distribution characteristics of the area, if the change rate of grayscale value can be analyzed while specifically suppressing the distribution through the radial direction, the interference of the hole outlet will be eliminated.

The Laplacian transform is a convolutional process with the Laplacian operator, a common method used in image processing. After analyzing the second-order differential distribution characteristics of grayscale values using the Laplacian operator, it is easy to distinguish images with different grayscale change rates. Laplacian transform is generally used to enhance image sharpness and effectively enhance edge features [36]. In this study, a filtering method based on the radial penalty Laplacian (RPL) operator is proposed for the visual measurement of deep holes in composite material components. This method can avoid the impact of radial trends in the grayscale value on the result while analyzing the second-order differentiation characteristics of the assembly hole image. For each pixel, two of its eight neighbors are along the radial direction. First, the image is divided into eight sector areas along the image center, as shown in Figure 7a. In each sector, a 3 × 3 convolution kernel is used as the Laplacian operator, and the two members representing the radial direction in the convolution kernel are determined according to the location of the sector. As shown in Figure 7b, the values of the corresponding convolution kernel members along the radial direction are set to 0, where the values of the other convolution kernel members are set to −1; thus, the Laplacian operator is determined in each sector.

As shown in Figure 8, the image of the assembly hole is divided into eight parts according to the corresponding sectors, and each part is convolved with its own Laplacian operator. Then, the processing results are stitched back together. Using this method, the impact of the grayscale value change in the radial direction on the results of the second-order differential analysis can be suppressed.

After using the RPL operator, the distribution of grayscale values along the radial direction around the hole outlet edge image is shown in Figure 9a. Pixels on both areas of the hole center and the hole inner wall are evenly distributed, and the original mutation point at the hole outlet edge has been removed. Figure 9b shows the distribution of grayscale values along the radial direction around the hole inlet edge. On one side of the hole inlet edge on the workpiece surface, the grayscale value change rate increases, whereas the grayscale values on the hole’s inner wall are suppressed close to 0. A mutation point of grayscale distribution is clearly generated on the inlet edge. Therefore, the image processing of the RPL operator meets the requirements of deep holes in the composite material components. However, before the hole edge can be extracted based on the grayscale distribution, smoothing methods such as dilation and median filtering are still necessary.

## 3. Deep Hole Measurement Method on the Composite Material Surface

The visual measurement of deep holes in composite workpieces involves three stages: extracting the regions of interest (ROIs), identifying the hole inlet edge, and fitting a circular edge. The measurement procedure is illustrated in Figure 10.

### 3.1. Extract the ROI

During the robotic assembly of aerospace components, the camera is required to have a wide measurement range to ensure that the target hole is in the measurement field of the camera. Consequently, the assembly hole occupies only a small part of the image. An ROI extraction algorithm is required to obtain a smaller image containing the assembly hole, reducing the amount of data while retaining effective information.

Neural networks use multi-layer convolution to extract image features and introduce pooling layers and convolutional layers in the network architecture to control the results of each layer’s convolution and classification. The YOLO neural network integrates features and classification into a single network and can output the target information detected simultaneously, including the category and location. In addition, the detection speed of the YOLO neural network can satisfy the requirement for real-time measurement systems [37]. In this study, a YOLOv3 neural network is used to segment the ROI with the assembly hole in the image.

The YOLO neural network achieves end-to-end object detection through convolutional operations. The image is divided into S × S grids, and the objects falling into each grid are detected. Finally, the bounding box, localization, and the confidence degree of all targets are given. The ROI extraction process based on the YOLO neural network includes the following steps: (1) converting the input image to the fixed size of 416 × 416 by employing a downsampling method; (2) loading the converted image into the network to obtain the detection results; (3) using the non-maximum suppression algorithm to remove the redundant targets; and (4) remapping the bounding box back onto the source image and segmenting the image according to the boundaries. The proposed ROI extraction process is accomplished via cropping the original image based on the bounding box, which maintains the clarity of the original image without a loss in resolution. The network structure of YOLOv3 is shown in Figure 11, using Darket-53 as the backbone extraction network and three different sizes of feature maps to detect objects of different sizes.

The cost function of the YOLOv3 neural network is shown in Equation (1):(1)$$\begin{array}{ll} L_{YOLO}= & \lambda_{box}\sum_{i=0}^{N\times N}\sum_{j=0}^{3}1_{ij}^{obj}[(t_{x}-t_{x}^{’}{)}^{2}+(t_{y}-t_{y}^{’}{)}^{2}]\\ & +\lambda_{box}\sum_{i=0}^{N\times N}\sum_{j=0}^{3}1_{ij}^{obj}[(t_{w}-t_{w}^{’}{)}^{2}+(t_{h}-t_{h}^{’}{)}^{2}]\\ & -\lambda_{obj}\sum_{i=0}^{N\times N}\sum_{j=0}^{3}1_{ij}^{obj}log(c_{ij})\\ & -\lambda_{nobj}\sum_{i=0}^{N\times N}\sum_{j=0}^{3}1_{ij}^{nobj}log(1-c_{ij})\\ & -\lambda_{class}\sum_{i=0}^{N\times N}\sum_{j=0}^{3}1_{ij}^{obj}\underset{c\in classes}{\sum }[p_{ij}^{’}(c)log(p_{ij}(c))\\ & +(1-p_{ij}^{’}(c))\log(1-p_{ij}(c))] \end{array}$$

In Equation (1), *t_x_*, *t_y_*, *t_w_*, and *t_h_* are the offsets between the ROI box of the assembly hole and the anchor point, and *t_x_*^′^, *t_y_*^′^, *t_w_*^′^, and *t_h_*^′^ are the predicted offsets between the ROI box of assembly hole and the anchor point [38]. A dataset for training the neural network is created by taking pictures of assembly holes on the composite material workpiece. It is expanded by translating, rotating, and scaling the original images.

### 3.2. Identify the Hole Inlet Edge

After obtaining the ROI of the hole with the neural network, the assembly hole is located near the center of the image, which is suitable for applying RPL transformation. The process of hole inlet edge identification is shown in Figure 12. First, the radial penalty Laplace operator is used to enhance the hole inlet edge and remove the hole outlet edge. Then, dilation is used to fill in the small uneven patterns on the workpiece surface and eliminate small particle noise within the hole. The uneven patterns on the workpiece surface are similar to salt-and-pepper noise. Therefore, they can be filtered using median filtering while protecting the image edges from blurring. Binary operation is required with image grayscale transformation, using Logarithmic transformation of grayscale values, as shown in Equation (2).
*s* = *a* + *b* log (1 + *r*)(2)
where *r* is the grayscale value of the original image, and *s* is the value after logarithmic transformation. Image binarization is achieved by logarithmic transformation, which expands the pixels with lower values and compresses those with higher values. Subsequently, an edge detection algorithm is used to extract the closed edges as the hole inlet edge.

### 3.3. Inflection Point Removal and Hole-Fitting

A continuous contour around the hole inlet edge is obtained using the image second-order differential analysis and filtering process shown in Figure 12. The hole inlet edge is usually a circle or an ellipse, with the curvature changing continuously and smoothly [39]. However, due to noise interference, the identified hole edge still contains non-smooth inflection points and outliers deviating from the true hole inlet edge. To ensure the fitting accuracy, further noise removal is necessary. The Random Sample Consensus (RANSAC) algorithm is used to detect and remove the points that significantly deviate from the circular contour while obtaining the pseudo center of the point set, as shown in Figure 13.

Inflection points have a discontinuous curvature. By removing points near the inflection points, the point set can be smoothed. After using the RANSAC algorithm, a circle is fitted to obtain an initial pseudo-center, denoted as (*x*_1_, *y*_1_). As shown in Figure 14, a point p*_i_* is selected on the hole edge, and then another point p*_i+n_* is located on the edge with an interval of *n* points. The segment between p*_i_*and p*_i+n_* is used to fit a part of the edge where these *n* points are located. Using this approach, the entire circle contour can be fitted into a sequence of segments. Each of the segments with the slope *k*_0_ is close to a chord on the circle, which satisfies certain geometric rules. The line connecting the circle center and the midpoint of the chord is perpendicular to the chord itself. The position of the chord midpoint is obtained by the fitted line segments, denoted as (*x*_1_, *y*_1_). As shown in Figure 14, the theoretical slope *k* of the chord satisfies Equation (3).
(3)$$\left\{\begin{array}{l} \tan\theta =\frac{y_{2}-y_{1}}{x_{2}-x_{1}}\\ k=-\cot\theta =\frac{x_{1}-x_{2}}{y_{2}-y_{1}} \end{array}\right.$$

The theoretical slope *k* of the chord is obtained by Equation (3). As shown in Figure 14, whether the segment between p*_i_* and p*_i+n_* is an inflection point set is determined by the inconsistency between *k*_0_ and *k*. A threshold *k*th is given. If the slope *k*_0_ of a fitted line segment does not satisfy Equation (4), the points on the line will be removed because inflection points exist within them. Further, the precision and calculation time of the inflection point detection can be controlled by adjusting the value of *n*. The higher the value given, the higher the precision of the detection and the longer the calculation duration.
*k*_0_ ∈ [*k* − *k*_th_, *k* + *k*_th_](4)

A circular contour is fitted from the identified hole edge using the least square method, and the result shown in the original image is shown in Figure 15.

## 4. Experiments and Results

### 4.1. Experimental Setup

To verify the proposed visual measurement method, a test platform was designed to simulate the visual measurement process of the assembly hole on the composite component. The selected camera, acA2440-20gm (Basler, Ahrensburg, Germany), with a resolution of 2448 × 2048 pixels, was used with a fixed focal lens, which balances measurement accuracy and the transmission rate. Considering that the composite workpiece had a dark surface, a white light source was chosen to increase the contrast of the hole edge and improve the hole edge detection accuracy. The measured holes were on the surface of a carbon fiber-reinforced polymer (CFRP) workpiece, which had a diameter of φ 10 mm and a depth of 50 mm. The workpiece was suspended to simulate the external environmental light illuminated through the hole center in real scenarios.

The experimental platform, as shown in Figure 16, incorporates a camera holder that enables independent adjustment of the camera’s pitch, roll, and height via its various joints. During the construction of the experimental platform, the camera was initially mounted at an approximate position on the holder, with the workpiece subsequently placed on the platform. Fine-tuning of the focus of the camera lens and the position of the workpiece was performed to ensure that the camera accurately focused on the hole of the workpiece surface. Images of the hole were captured using the camera, and the image scale grids were used to visually evaluate if the outline of the tested hole conformed to a perfect circle. Manual micro-adjustments to the camera’s pitch and roll were then performed until the tested hole was displayed as a perfect circle. Thereafter, all joints of the camera holder were tightened to complete the camera’s installation. The camera was situated roughly 100 mm from the workpiece surface, with its optical axis accurately parallel to the target hole axis. Images of holes on the workpiece were captured utilizing the constructed measurement platform, and the holes were measured using the proposed method.

### 4.2. Experimental Results

The proposed RPL algorithm was tested on a dataset consisting of 20 images featuring deep holes in a composite material workpiece to validate its capability of noise suppression and edge enhancement. An improved texture segmentation algorithm based on Local Exponential Patterns (LEP) was also reproduced and compared against the RPL algorithm. Following the ROI extraction, the original images were separately processed using the RPL algorithm and the LEP algorithm, followed by a sequential image preprocess consisting of dilation, median filter, and binarization. Subsequently, the edge of the holes was extracted, as shown in Figure 17. When confronted with the challenges of image processing for deep holes in the composite material workpiece, the RPL algorithm effectively suppressed both the fine-grained noise introduced by surface patterns and the smoothing noise caused by halo effects around the hole outlet, exhibiting a superior performance compared to the LEP algorithm. The test was conducted on a computer powered by an AMD 5700 G CPU with 32 GB of RAM. Under these computational conditions, the RPL algorithm achieved an average processing time of 0.94 s per hole, unlike the LEP algorithm, which required 3.02 s per hole on average, as illustrated in Figure 17.

Furthermore, the performance of the proposed hole measurement algorithm was comprehensively tested by extracting the deep hole edges on the composite material surface. The Hough Circle Transform [16] and a semi-supervised deep learning approach [32] were utilized to serve as comparisons for evaluating the effectiveness of the proposed approach. This deep learning model, employing the U-Net network architecture, was trained on a dataset of hole images labeled using the LEP algorithm combined with a voting-ellipse-fitting method [26]. The original images of the training dataset were captured from the composite material deep holes via the proposed experimental platform, augmented with random rotations and random shifts in color channels applied to the initial images. The measurement results of various methods are shown in Figure 18. The Intersection over Union (IoU) and Dice indexes are widely used as metrics for assessing the performance of image segmentation algorithms [40].
IoU = |*P*_r_ ∩ *P*_m_| / |*P*_r_ ∪ *P*_m_|(5)
Dice = 2 × |*P*_r_ ∩ *P*_m_| / (|*P*_r_|+|*P*_m_|)(6)
where *P*_r_ is the pixels constituting the actual hole area, and *P*_m_ denotes the pixels encompassed by the extracted hole edge. The IoU and Dice indexes under various measurement methods of 20 hole images are shown in Table 1. Regarding the visual measurement of deep holes in composite materials, the proposed method demonstrates superior accuracy. The execution time for a single-hole measurement using the proposed method was recorded as 2.73 s.

### 4.3. Measurement Accuracy Verification

Visual measurement of the assembly holes on the composite material workpiece was performed to obtain the position and radius of holes, and it is necessary to convert the values of the radius or hole position into a physical coordinate system from the pixel coordinate system. To verify the measurement accuracy of the proposed algorithm, a coordinate measuring machine (CMM) (PRISMO7S-ACC, Zeiss, Oberkochen, Germany) was used for comparison measurement, as shown in Figure 19. Two groups of holes on a CFRP workpiece were measured; each group contained two adjacent holes. The measurement results are listed in Table 2.

The imaging using an industrial camera with a fixed focal lens follows the principle of pinhole imaging. According to the perspective model, the transformation of a point P from the pixel coordinate system (*x*, *y*) to the physical coordinate system (*X*_c_, *Y*_c_, *Z*_c_) can be expressed as Equation (7):(7)$$\left\{\begin{array}{c} \frac{x}{f}=\frac{X_{c}}{Z_{c}}\\ \frac{y}{f}=\frac{Y_{c}}{Z_{c}} \end{array}\right.$$
where *f* denotes the lens focal length. For two adjacent holes in an image, the distance between them in the pixel coordinate system is denoted as *l*_pix_, which follows Equation (8):(8)$$l=\frac{Z_{c}}{f}l_{pix}$$
where *l* is the Euclidean distance between two holes, and *Z*_c_/*f* represents a scaling factor used to achieve conversion between the pixel coordinate system and the physical coordinate system.

The visual measurement system in the experiment can measure two holes in the same field, as shown in Figure 20. Using the proposed hole measurement method, the center location and radius of the two adjacent holes were obtained in the pixel coordinate system, and their center distance *l*_pix_ was calculated.

While *l* is given in Table 2, the scaling factor *Z*_c_/*f* was calculated according to Equation (8), which is 44.0279. Further, the radius of the measured hole can be calculated as
(9)$$r=\frac{Z_{c}}{f}r_{pix}$$

The radius of the four holes on the CFRP workpiece using the proposed visual measurement method is shown in Table 3, which also presents the comparison measurement results attained using the CMM. According to the comparison results, the radius error of visual measurement is less than 0.03 mm.

## 5. Conclusions

In this study, a method for visual measurement of deep holes in a composite material workpiece is proposed to solve the interference from the composite material surface pattern, as well as the nested edge problem from the inlet and outlet edges of a deep hole in the image. An image processing method with the RPL operator was designed to eliminate the outlet edge of the hole while enhancing the inlet edge utilizing the radial distribution characteristics of grayscale values. The point set of the hole inlet edge was smoothed by removing the inflection points to fit a more accurate hole edge. Additionally, to improve the real-time performance and reliability of the algorithm, a neural network was added to extract the region of interest containing assembly holes for the subsequent edge enhancement algorithm and hole measurement process. The experimental results illustrate the following:(1)The RPL algorithm demonstrates substantial effectiveness in the image processing of deep holes on a composite material surface, effectively mitigating surface texture-related and smoothing noise caused by halo effects at the hole outlet.(2)The proposed inflection point removal and hole fitting method effectively removes noise from the contour point set, ensuring the precise extraction of the hole edge.(3)The comparison experimental results validated by the Coordinate Measuring Machine (CMM) verify that the visual measurement accuracy of the hole size reaches 0.03 mm, which meets the accuracy requirement in the aerospace component assembly process.

It should be noted that the proposed method is still not suited for elliptical contours arising from tilted hole measurement or those on a curved surface; moreover, the measurement time of 2.73 s per hole proves sluggish for efficient hole detection operations. Further study and improvement are still needed to resolve these aforementioned limitations.

## Figures and Tables

**Figure 1 sensors-24-03786-f001:**
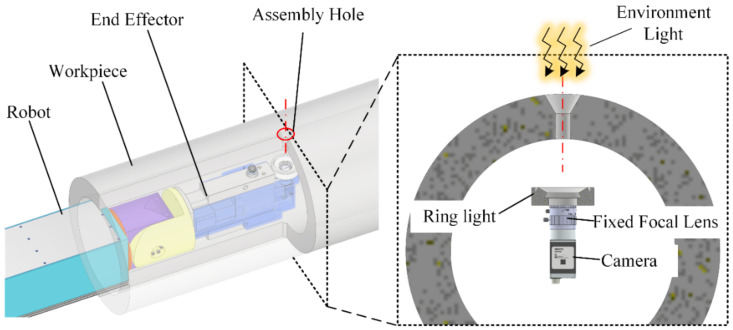
Visual measurement of deep holes in the cavity-shaped composite material workpiece.

**Figure 2 sensors-24-03786-f002:**
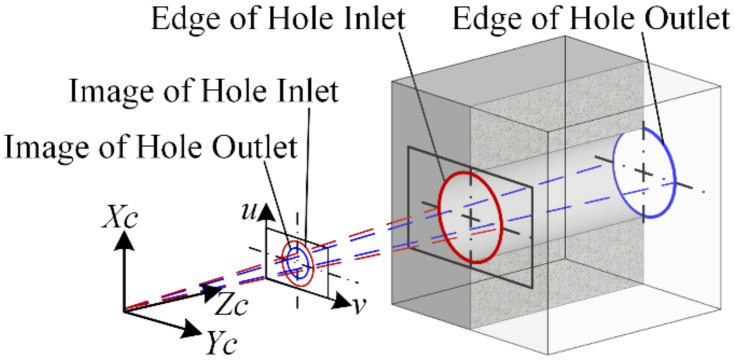
Imaging characteristics of deep holes based on the pinhole perspective model.

**Figure 3 sensors-24-03786-f003:**
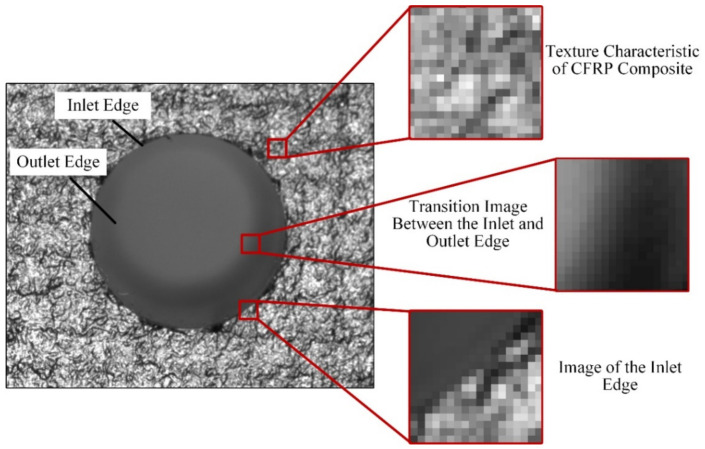
Local feature analysis of visual imaging for deep holes in the composite material surface.

**Figure 4 sensors-24-03786-f004:**
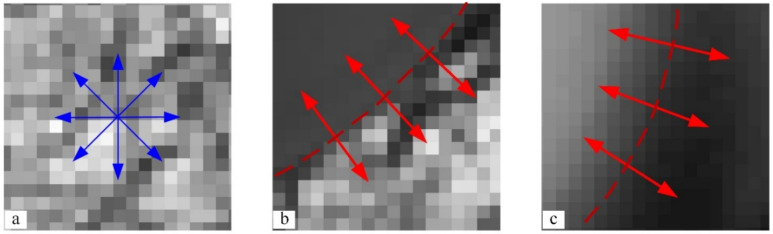
Three typical local distributions of grayscale values in assembly hole imaging. (**a**) The composite material surface with the grayscale changing sharply in each direction, as indicated by the blue arrows; (**b**) the hole inlet edge with the grayscale changing sharply only in the radial direction, as highlighted by the red arrows; (**c**) the hole outlet edge with the grayscale changing gradually in the radial direction, as highlighted by the red arrows.

**Figure 5 sensors-24-03786-f005:**
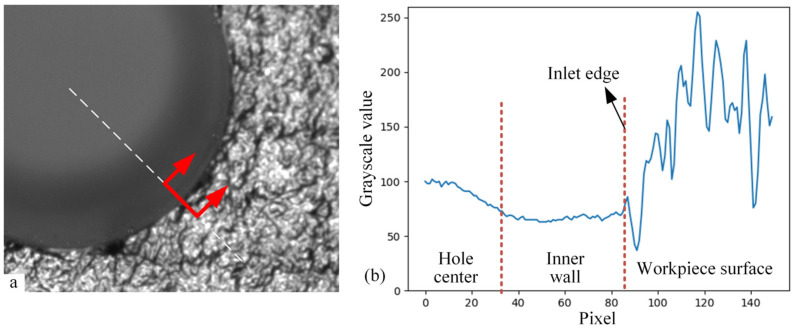
Radial distribution of grayscale values around the hole inlet edge. (**a**) Sample selected from the original image along the radial direction around the hole inlet edge, as highlighted between the red arrows; (**b**) the one-dimensional pixel distribution with a significant difference in the grayscale change rate at the inlet edge.

**Figure 6 sensors-24-03786-f006:**
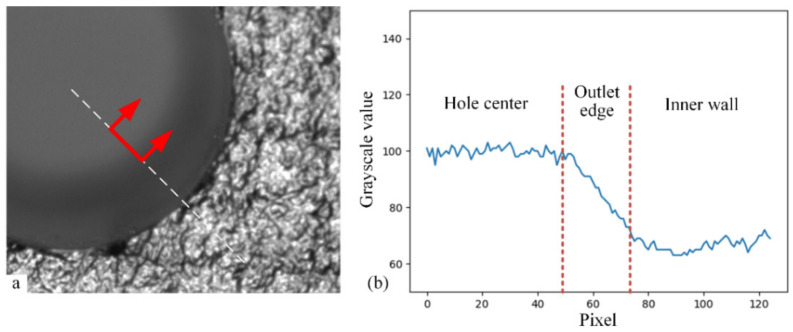
Radial distribution of grayscale values around the hole outlet edge. (**a**) Sample selected from the original image along the radial direction around the hole outlet edge, as highlighted between the red arrows; (**b**) the one-dimensional pixel distribution with a moderate grayscale changing through the outlet edge.

**Figure 7 sensors-24-03786-f007:**
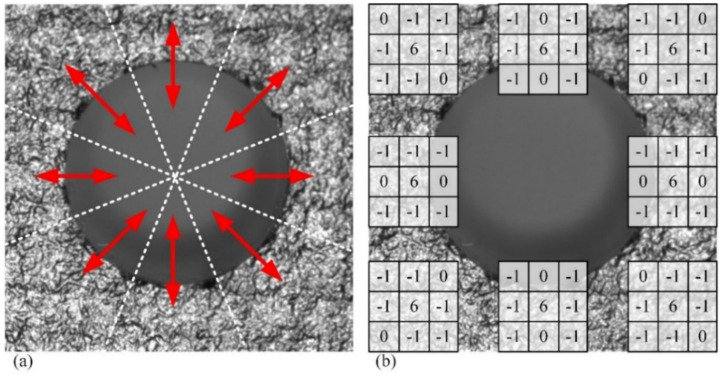
Sectorization of images and RPL operator for each sector. (**a**) The eight sectors are divided evenly from the image according to the central symmetry principle, with the red arrows indicating the radial principal direction of each sector; (**b**) 3 × 3 Laplacian operator for each sector with the members along the radial direction set as 0.

**Figure 8 sensors-24-03786-f008:**
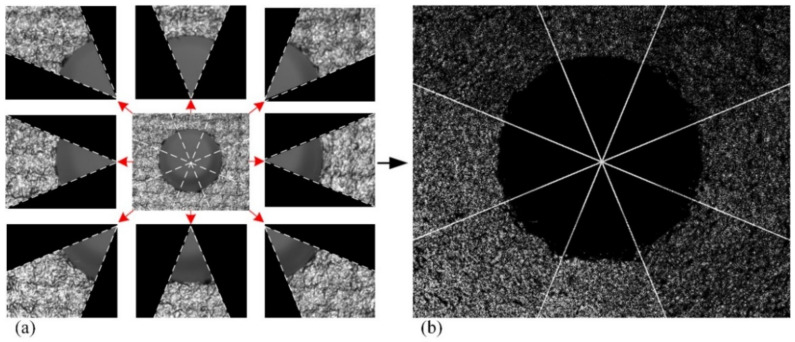
The effect of sectorization RPL transform. (**a**) Each sector filled out as rectangles for Laplacian transform. (**b**) The stitched image after processing.

**Figure 9 sensors-24-03786-f009:**
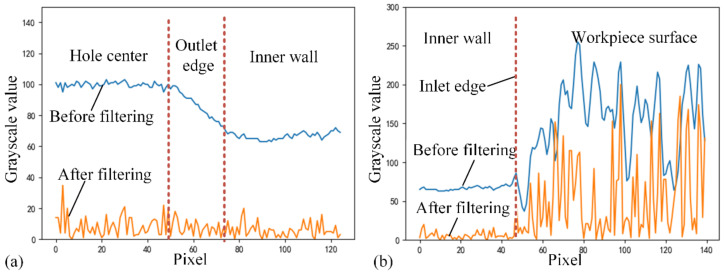
One-dimensional radial pixel distribution before and after RPL transform. (**a**) The hole outlet features are suppressed well; (**b**) the hole inlet features are enhanced.

**Figure 10 sensors-24-03786-f010:**
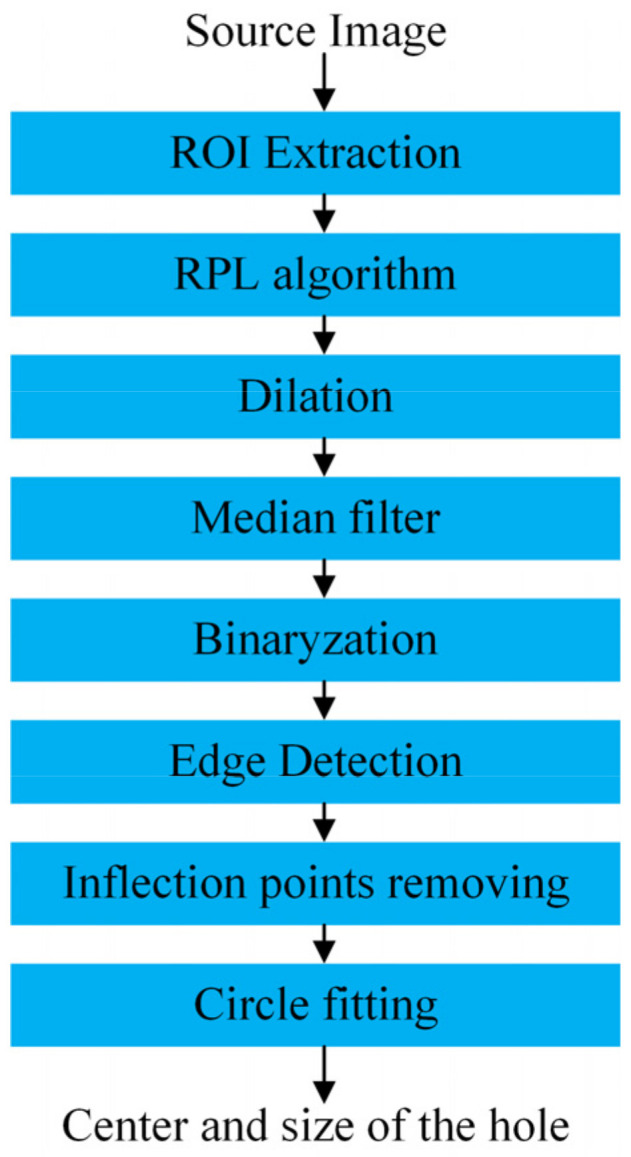
The measurement procedure of deep holes in the composite material surface.

**Figure 11 sensors-24-03786-f011:**
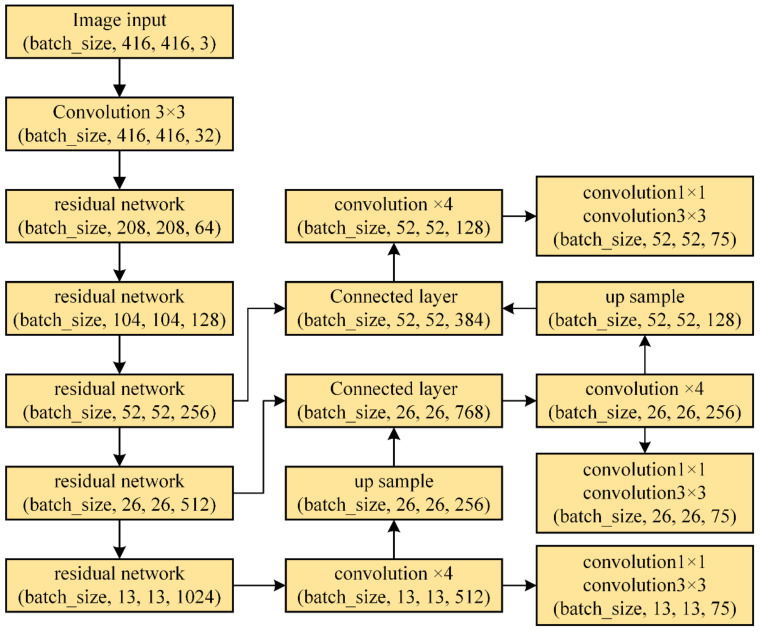
YOLOv3 neural network structure.

**Figure 12 sensors-24-03786-f012:**
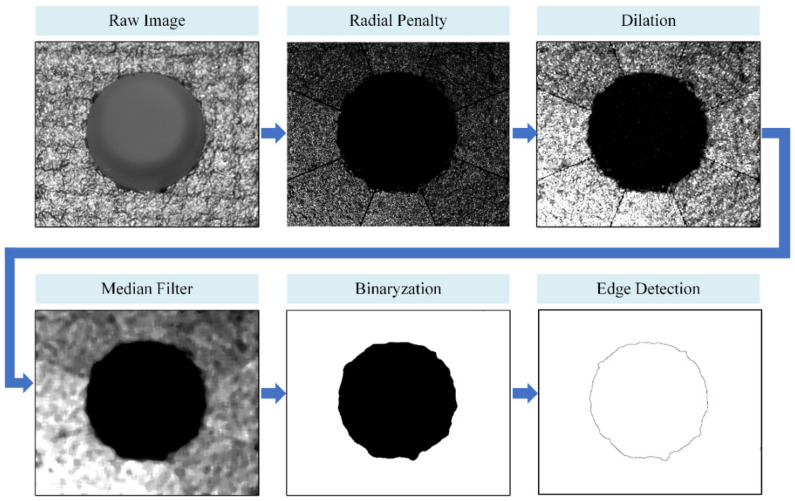
Image processing for identifying the hole inlet edge.

**Figure 13 sensors-24-03786-f013:**
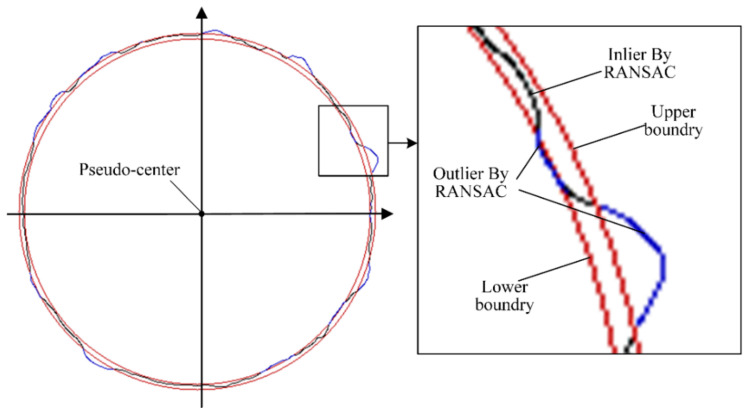
A RANSAC-based method for pseudo-center extraction and outlier removal.

**Figure 14 sensors-24-03786-f014:**
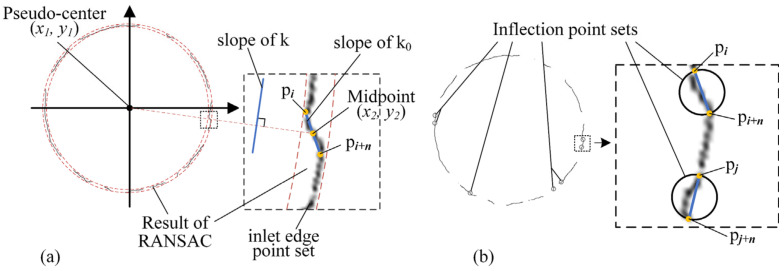
An inflection point searching method based on the regularity of slope distribution. (**a**) By selecting two points, denoted as p*_i_* and p*_i+n_*, from the result of RANSAC with an interval of *n*, the line passing through the two points has a slope of *k*_0_. Meanwhile, the perpendicular direction of the line passing through both the pseudo center and the midpoint of p*_i_* and p*_i+n_* has a slope of *k*. (**b**) Inflection point sets searched from the hole edge.

**Figure 15 sensors-24-03786-f015:**
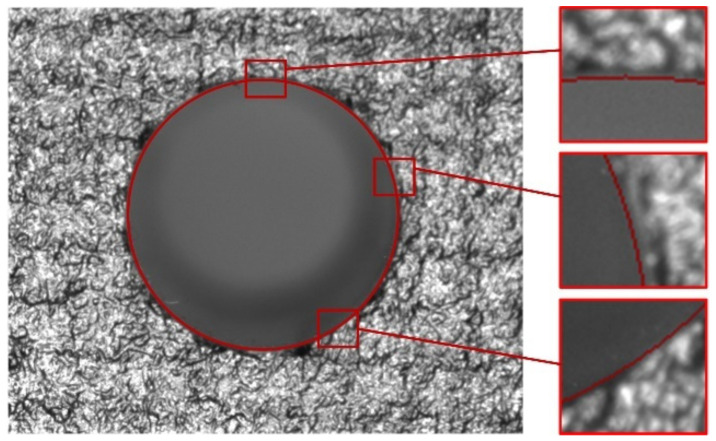
Assembly hole measurement result.

**Figure 16 sensors-24-03786-f016:**
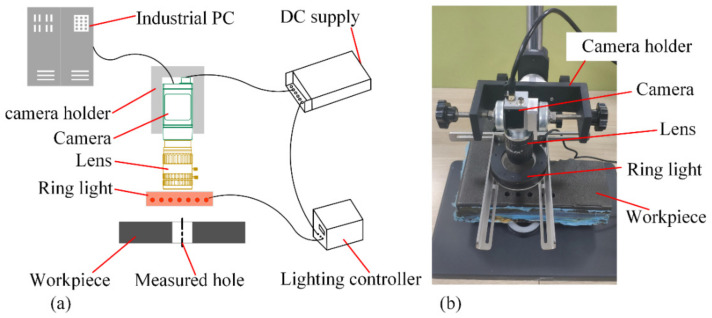
Visual measurement experiment platform for measuring deep holes on a composite material surface. (**a**) Overall design of the experimental platform; (**b**) the visual measurement site.

**Figure 17 sensors-24-03786-f017:**
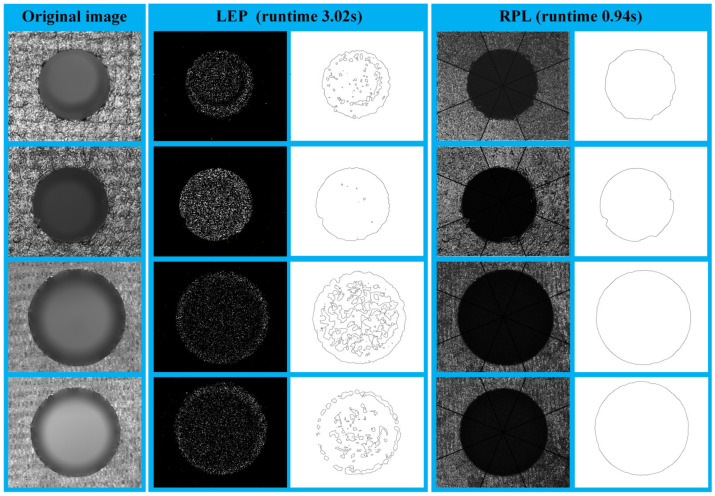
Comparison results of LEP and the proposed RPL.

**Figure 18 sensors-24-03786-f018:**
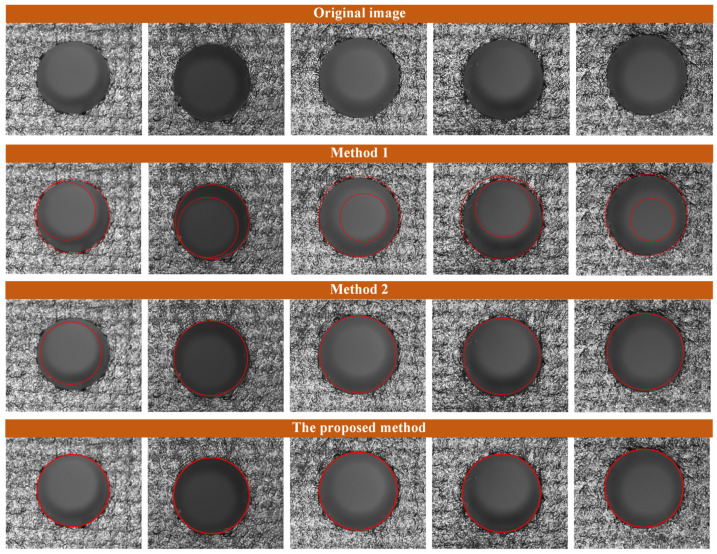
Visual measurement results of deep holes in CFRP workpiece. Method 1 utilizes the Hough Circle Transform; Method 2 utilizes the U-Net network model.

**Figure 19 sensors-24-03786-f019:**
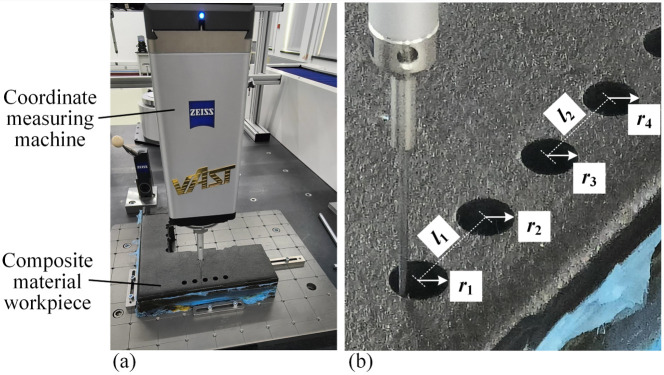
Measurement of holes on a composite material workpiece with the CMM. (**a**) The measurement site; (**b**) the parameters to be measured.

**Figure 20 sensors-24-03786-f020:**
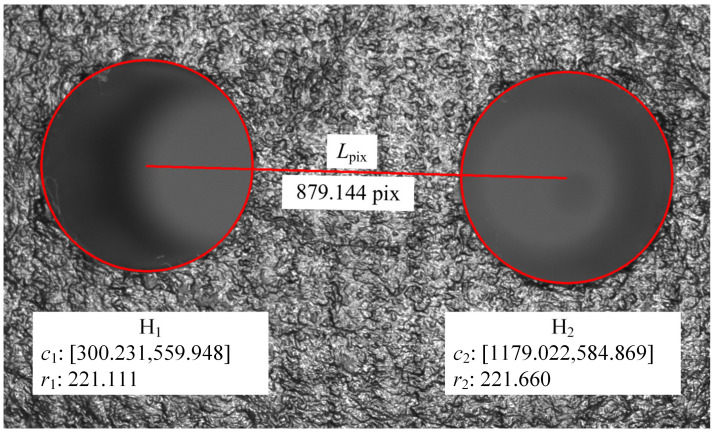
Visual measurement results of two adjacent holes.

**Table 1 sensors-24-03786-t001:** IoU and Dice index under different methods of hole measurement.

Method	IoU	Dice
Hough Circle Transform [16]	0.6485	0.7868
U-Net with LEP semi-supervised [32]	0.9020	0.9485
The proposed method	0.9695	0.9845

**Table 2 sensors-24-03786-t002:** Measurement results of the CMM.

*r*_1_ (mm)	*r*_2_ (mm)	*r*_3_ (mm)	*r*_4_ (mm)	*l*_1_ (mm)	*l*_2_ (mm)
5.055	5.053	5.046	5.043	19.998	19.954

**Table 3 sensors-24-03786-t003:** Visual measurement accuracy.

	Visual Measurement	CMM	Visual Measurement Error
*r* _1_	5.035 mm	5.055 mm	0.020 mm
*r* _2_	5.024 mm	5.053 mm	0.029 mm
*r* _3_	5.025 mm	5.046 mm	0.021 mm
*r* _4_	5.026 mm	5.043 mm	0.017 mm

## Data Availability

The dataset is available from the authors upon request.

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
