# Peer review of "A Visual Measurement Method for Deep Holes in Composite Material Aerospace Components"

_sensors, 2024, doi:10.3390/s24123786_

Round 1
Reviewer 1 Report
Comments and Suggestions for Authors
Due to the following concerns and as a result of opaque technical contributions in science border, the current submission cannot be recommended.
It suffers from poor and shallow technical Introduction with trivial old references. Overall, doesn’t clearly show, the main highlighted research gaps, not critically analyzed the most relevant and updated works, the main characterized worthwhile novelty over previous works, … Definitely MUST be decorated with in-depth review and updated work mostly from the last 5 years ago.
Overall,
First: As far as can be understood, the prerequisite condition of this work is the use of unified pixel size, and therefore it cannot be functional for images with different resolutions and thus based on the used datasets it can only be acceptable when the inputs don’t change, else in the case of modified images it will no longer match and hence, the integrity cannot be verified. Your solution??
Second: the submitted draft doesn’t have any certified and documented Discussion in terms of accuracy metrics, evidential analysis, solid comparison with other scholars, limitation, stability approval, uncertainty quantifications, pitfall and practical difficulties, physical interpretation, comparison with ground truth data, computational time and cost based on the system used, noise reduction and image quality, treating the occlusions and partial visibility, multiple objects, overlapping regions, the impact of the bias of the used dataset on current feature extracting system…
Third: the lack of validation and model optimality based on the hyperparameter optimizations cannot be neglected.
Fourth: predictability, calibrating and sensitivity analysis should be carried out through the weight database of the trained model to show the importance of the used attributes when its optimality is confirmed. Clarify where and how the optimal weight database is stored? How it can be recalled?? Search for updating the neural network models using different sensitivity analysis methods, sensitivity analysis for neural networks, novel feature selection using sensitivity analysis…
In addition, the authors are encouraged to address the following concerns:
1. Highlight the 1-A) main research gaps and the reason why you are motivated for this work, 1-B) outline the main innovations in a concise form, 1-C) Strongly recommended for critically analyzed updated relevant works
2) Concerning the applied model and used datasets:
2-1) Source of the datasets?? Their resolutions?
2-2) The gathered datasets in the field of AI must be validated using benchmarks and therefore, documenting the work in the current format cannot be certified. This means that any captured image can be considered as new data where definitely is not correct.
2-3) Whether you are building inference/predictive model, you will achieve better results by first verifying that you have chosen the optimal set of features to train your model with. Therefore, getting better results obviously is not a big deal.
2-4) your method in Fig 10, somewhat follows the task decomposition which can be executed using other advanced approaches using block-based or modular structures. Search for keywords like prediction using a block combined network structure; mapping using hybridized block modular model, image denoising using 3D filtering and block matching CNN, block-based CNN for image forgery detection, … and enrich the literature by giving the advantage of your work over them.
3) Conclusion could be justified much better.
Comments on the Quality of English Language1. Rigor English proofread with the help of a native agent.
Author Response
For research article
|
Response to Reviewer 1 Comments
|
||
|
1. Summary |
|
|
|
Thank you very much for taking the time to review this manuscript. Please find the detailed responses below and the corresponding revisions/corrections highlighted/in track changes in the re-submitted files.
|
||
|
2. Questions for General Evaluation |
Reviewer’s Evaluation |
Response and Revisions |
|
Does the introduction provide sufficient background and include all relevant references? |
Must be improved |
The Introduction has undergone substantial expansion and updating, incorporating research advancements in the last five years. It presents a meticulously revised overview of the current research, and includes targeted, critical analysis that is specifically aligned with the challenges confronted in this study. |
|
Are all the cited references relevant to the research? |
Must be improved |
More references from the last five years have been cited. References with weak relevance have been removed. |
|
Is the research design appropriate? |
Must be improved |
The experimental methods have been refined, employing more objective approaches to analyze and validate the proposed method. |
|
Are the methods adequately described? |
Must be improved |
The research methodologies employed have been described in greater detail, with the inclusion of new flowchart to illustrate the entire process. |
|
Are the results clearly presented? |
Must be improved |
The experimental results have been re-written, incorporating a larger dataset, with more sophisticated method reproduced and compared. A more impartial algorithm characterization methodology has been adopted. |
|
Are the conclusions supported by the results? |
Must be improved |
The conclusions of the study have been reorganized and restated in a point-by-point format. |
|
3. Point-by-point response to Comments and Suggestions for Authors |
||
|
Comments 1: It suffers from poor and shallow technical Introduction with trivial old references. Overall, doesn’t clearly show, the main highlighted research gaps, not critically analyzed the most relevant and updated works, the main characterized worthwhile novelty over previous works, … Definitely MUST be decorated with in-depth review and updated work mostly from the last 5 years ago. |
||
|
Response 1: Thank you for your suggestion. The authors fully concur with the reviewer's astute observation regarding the inadequacy in the technical introduction and has conducted an exhaustive review to further supplement the manuscript with recent advancements in hole visual measurement research from the last five years, exemplified by references [20], [30-32], among others. The authors have also gone through the cited works again, try to emphasize their key innovative ideas. Moreover, a critical appraisal has been undertaken, focusing on the challenges pertinent to this study, leading to a refined and more comprehensive analysis. These enhancements aspire to address the reviewer's concerns effectively and enrich the manuscript's content. The modifications may be found in the revised manuscript at lines 61-65, 75-80, 103-107, and line 114. |
||
|
Comments 2: As far as can be understood, the prerequisite condition of this work is the use of unified pixel size, and therefore it cannot be functional for images with different resolutions and thus based on the used datasets it can only be acceptable when the inputs don’t change, else in the case of modified images it will no longer match and hence, the integrity cannot be verified. Your solution?? |
||
|
Response 2: We apologize for the confusion caused to the reviewer. Firstly, it is important to explain that the proposed algorithm itself does not impose limitations on the size of input images. However, during the Regions of Interest (ROI) extraction process, the input images are restricted to a size of 416 × 416 due to the YOLO v3 model. According to the authors' practical design, at the beginning of the ROI extraction, the input images will be downsampled and resized. And at the conclusion of the ROI extraction process, bounding boxes are remapped to the coordinate system of the input image to accomplish image segmentation. Explanation of this procedure is also provided in lines 261-265 of the revised manuscript. |
||
|
Comments 3: The submitted draft doesn’t have any certified and documented Discussion in terms of accuracy metrics, evidential analysis, solid comparison with other scholars, limitation, stability approval, uncertainty quantifications, pitfall and practical difficulties, physical interpretation, comparison with ground truth data, computational time and cost based on the system used, noise reduction and image quality, treating the occlusions and partial visibility, multiple objects, overlapping regions, the impact of the bias of the used dataset on current feature extracting system… |
||
|
Response 3: Thank you for your valuable suggestion. The authors, in accordance with the suggestions, have reconceived the experimental validation section in Chapter 4. A semi-supervised deep learning approach, previously proposed for hole detection in composite materials, was reproduce and included it in this study as a point of comparison (lines 374); A new dataset comprising 20 images of different deep holes on composite material workpiece was captured to validate the performance of the proposed algorithm, with a particular focus on comparing its capabilities in noise reduction and image features enhancement (lines 361). Runtime of the proposed method for measuring a single hole was repeated tested as 2.73s on specific computational platform (lines 393). The Intersection over Union (IoU) and Dice index for hole measurements by various methods were calculated, serving as metrics to evaluate and compare their performance (table 1). In accordance with the recommendations, the authors have also rewritten the conclusion, summarized the limitations of this study and outlined directions for further improvement (lines 455-458). Nonetheless, what the author still needs to explain to the reviewer is that the primary objective of this research is to tackle the visual measurement challenges of deep holes on composite material workpiece. Thus, the proposed method currently does not address complications arising from occlusions, multiple objects or overlapping, and the collected dataset, being rather specialized, lacks the generality required for hole measurements across diverse scenarios. |
||
|
Comments 4: the lack of validation and model optimality based on the hyperparameter optimizations cannot be neglected |
||
|
Response 4: We apologize for the confusion caused to the reviewer. The proposed algorithm constitutes an improvement on the Laplacian transform, and the method for identifying holes remains rooted in conventional image processing techniques. Within such conventional image processing algorithms, the notion of hyperparameter optimization is less emphasized compared to its prominence in machine learning algorithms. Based on the image characteristics of deep holes in composite materials and the principles underlying the proposed radially penalized Laplace transform, the size of the selected Laplacian operator and the values of its components are fixed. Throughout the comprehensive image processing workflow, the kernel size for the median filtering algorithm has undergone minor tuning through practical testing. Therefore, automatic optimization of model parameters was not carried out in this study. |
||
|
Comments 5: predictability, calibrating and sensitivity analysis should be carried out through the weight database of the trained model to show the importance of the used attributes when its optimality is confirmed. Clarify where and how the optimal weight database is stored? How it can be recalled?? Search for updating the neural network models using different sensitivity analysis methods, sensitivity analysis for neural networks, novel feature selection using sensitivity analysis… |
||
|
Response 5: We apologize for the confusion caused by the neural network structure in Fig.11. The YOLO v3 model was only employed in the ROI extraction process, while the proposed RPL image processing algorithm and hole measurement methodology remain grounded in conventional computer vision techniques. Nonetheless, the authors remain highly interested in the questions raised here, and look forward to exploring answers in future research endeavors. |
||
|
Comments 6: Highlight the 1-A) main research gaps and the reason why you are motivated for this work, 1-B) outline the main innovations in a concise form, 1-C) Strongly recommended for critically analyzed updated relevant works |
||
|
Response 6: Thank you for raising the suggestions. The authors are pleased to address each concern point by point. 1-A) The demand for visual measurement of deep holes on composite material workpiece addressed in this study arises from the research on robotic automated assembly techniques for composite skin panels of spacecraft sections. This process entails the robot drilling into the interior of the compartments, precisely measuring the positions of assembly holes, and assessing the dimensions of these holes, thereby enabling the inspection of manufacturing quality. The measurement challenges for deep holes on composite material workpiece stem from the combined interference of fine-grained noise introduced by surface patterns and the smoothing noise caused by halo effects around the hole outlet. A detailed account of the challenges inherent in this issue is also provide in lines 81-87 of the manuscript, along with the necessity for such measurement approach 1-B) Addressing the challenges of visual measurement for deep holes on composite material workpiece, this study analyzes the visual characteristics of the images and, based on these findings, creatively proposes the RPL (Radial Penalty Laplacian) algorithm for edge enhancement and noise suppression. It further develops a precise measurement methodology for composite material deep holes, including an advanced inflection points removal algorithm. 1-C) According to the reviewers' comments, the authors have conducted a deeper analysis of the current state of research in this field, incorporating more recent relevant publications. Focusing on the issues encountered in our study, we have scrutinized the shortcomings and inappropriateness of existing research content. The modifications may be found in the revised manuscript at lines 61-65, 75-80, and line 114. |
||
|
Comments 7: Concerning the applied model and used datasets: 2-1) Source of the datasets?? Their resolutions? 2-2) The gathered datasets in the field of AI must be validated using benchmarks and therefore, documenting the work in the current format cannot be certified. This means that any captured image can be considered as new data where definitely is not correct. 2-3) Whether you are building inference/predictive model, you will achieve better results by first verifying that you have chosen the optimal set of features to train your model with. Therefore, getting better results obviously is not a big deal. 2-4) your method in Fig 10, somewhat follows the task decomposition which can be executed using other advanced approaches using block-based or modular structures. Search for keywords like prediction using a block combined network structure; mapping using hybridized block modular model, image denoising using 3D filtering and block matching CNN, block-based CNN for image forgery detection, … and enrich the literature by giving the advantage of your work over them. |
||
|
Response 7: We apologize for the confusion caused to the reviewer. The authors are pleased to address each concern point by point. 2-1) The dataset employed in this study was compiled by the authors using a Basler acA2440-20gm camera to capture images of deep holes on a CFRP workpiece. All holes featured a consistent depth of 50mm and a diameter of 10mm. The dataset comprises 30 images, all of which are at a resolution of 2448×2048 pixels. 2-2) Although this research is centered on conventional computer vision algorithms, we fully acknowledge the importance of data validation in ensuring the reliability of our algorithms and the validity of our findings. Given the absence of publicly available, annotated reference datasets specific to the visual inspection of deep holes in composite materials, we have independently designed and implemented an experimental platform to collect image data tailored to this particular need. We recognize the significant value our collected dataset holds for advancing research in this field and fostering knowledge sharing. Consequently, as our study progresses, we commit to progressively completing the labeling and organization of this dataset, with the intention of eventually releasing it as a public resource at an appropriate time. 2-3) The main algorithms employed in this research, including the RPL algorithm and the inflecting point removing method, are not trained through deep learning models, but rather are based on traditional algorithms grounded in specific mathematical principles. Selection of parameters for the proposed algorithms primarily relies on the characteristics of the images and the underlying principles of the algorithms themselves, hence no extensive optimization measures for algorithm parameters have been undertaken. 2-4) Thanks for the reviewer's suggestions. The authors have carefully reviewed the keywords and associated accomplishments the reviewer provided, uncovering several outstanding hole recognition methods based on deep learning technologies, which have enriched my analysis of the current state of research. The Fig 10 illustrates the ROI (region of interest) extraction approach adopted in this study, utilizing YOLOv3, aimed at swiftly identifying and segmenting areas containing holes. The authors utilized their self-curated dataset to rapidly train this model, promptly yielding favorable outcomes. However, given that the ROI extraction process does not strive for (nor is capable of achieving) precise hole localization and diameter measurement, no further optimization or adjustments to this model were pursued. |
||
|
Comments 8: Conclusion could be justified much better. |
||
|
Response 8: Thank you for raising the suggestion. The authors have meticulously considered the reviewers' comments and have subsequently revised the summary, categorizing the existing content into distinct sections for clarity. Moreover, they have conducted a deeper analysis on the shortcomings of the present research and outlined potential avenues for future investigation. The modifications may be found in the revised manuscript at lines 445-458. |
||
|
4. Response to Comments on the Quality of English Language |
||
|
Point 1: Rigor English proofread with the help of a native agent. |
||
|
Response 1: Thanks for the suggestion. Authors fully comprehend the importance of English proficiency in meeting the publication standards of the journal, and have thoroughly examined the entire document to guarantee grammatical accuracy and term exactness. Before resubmitting the manuscript, the authors also intend to enlist the services of MDPI's language polishing to ensure its linguistic quality.
|
||
|
5. Additional clarifications |
||

Reviewer 2 Report
Comments and Suggestions for Authors
This paper proposed a visual measurement method for deep holes on composite materials based on the radial penalty Laplacian operator. The main contribution of this paper is a hole edge enhancement algorithm based on radial penalty laplacian operator. It is an interesting work. The proposed method sounds good and the results look promising.
The following are my concerns:
(1) It is not enough to only compare with traditional Hough methods. I would like to see more evaluation results with the SOTA methods, including deep learning methods.
(2) Does the performance improvements have statistical significance?
(3) For the performance evaluation, it is not sufficient without any performance comparisons. I would like to see the comparison results with traditional Hough methods and state-of-the-art methods.
(4) The limitations of the proposed method are missing.
(5) No relevant experiments to discuss the running time.
Comments on the Quality of English LanguageN/A
Author Response
For research article
|
Response to Reviewer 2 Comments
|
||
|
1. Summary |
|
|
|
Thank you very much for taking the time to review this manuscript. Please find the detailed responses below and the corresponding revisions/corrections highlighted/in track changes in the re-submitted files.
|
||
|
2. Questions for General Evaluation |
Reviewer’s Evaluation |
Response and Revisions |
|
Does the introduction provide sufficient background and include all relevant references? |
Can be improved |
The Introduction has undergone substantial expansion and updating, incorporating research advancements in the last five years. It presents a meticulously revised overview of the current research, and includes targeted, critical analysis that is specifically aligned with the challenges confronted in this study. |
|
Are all the cited references relevant to the research? |
Can be improved |
More references from the last five years have been cited. References with weak relevance have been removed. |
|
Is the research design appropriate? |
Can be improved |
The experimental methods have been refined, employing more objective approaches to analyze and validate the proposed method. |
|
Are the methods adequately described? |
Can be improved |
The research methodologies employed have been described in greater detail, with the inclusion of new flowchart to illustrate the entire process. |
|
Are the results clearly presented? |
Can be improved |
The experimental results have been re-written, incorporating a larger dataset, with more sophisticated method reproduced and compared. A more impartial algorithm characterization methodology has been adopted. |
|
Are the conclusions supported by the results? |
Can be improved |
The conclusions of the study have been reorganized and restated in a point-by-point format. |
|
3. Point-by-point response to Comments and Suggestions for Authors |
||
|
Comments 1: It is not enough to only compare with traditional Hough methods. I would like to see more evaluation results with the SOTA methods, including deep learning methods. |
||
|
Response 1: Thank you for your valuable suggestion. The authors, in accordance with the suggestions, have reconceived the experimental validation section in Chapter 4. An improved texture segmentation algorithm base on Local Exponential Patterns (LEP) was reproduced and compared against the proposed RPL algorithm (lines 364). A semi-supervised deep learning approach, previously proposed for hole detection in composite materials, was also reproduced and included in this study as a point of comparison (lines 374). |
||
|
Comments 2: Does the performance improvements have statistical significance? |
||
|
Response 2: We apologize for the confusion caused to the reviewer. Statistical tests were performed to confirm the significance of the observed performance improvements. 20 images of deep holes on composite material workpiece were processed using various measurement method, and the Intersection over Union (IoU) and Dice index for hole measurements by various methods were calculated, serving as metrics to evaluate and compare their performance (table 1). According to the experimental results, the proposed algorithm significantly enhanced the accuracy of hole measurement. |
||
|
Comments 3: For the performance evaluation, it is not sufficient without any performance comparisons. I would like to see the comparison results with traditional Hough methods and state-of-the-art methods. |
||
|
Response 3: Thank you for your valuable suggestion. The authors, in accordance with the suggestions, have reconceived the experimental validation section in Chapter 4. An improved texture segmentation algorithm base on Local Exponential Patterns (LEP) was reproduced and compared against the proposed RPL algorithm (lines 364). A semi-supervised deep learning approach, previously proposed for hole detection in composite materials, was also reproduce and included it in this study as a point of comparison (lines 374). |
||
|
Comments 4: The limitations of the proposed method are missing. |
||
|
Response 4: The limitations of the proposed method include: (1) the proposed method is still not suited for the issue of elliptical contours arising from tilt-ed hole measurement or those on curved surface; (2) the measurement time of 2.73 seconds per hole proves sluggish for efficient hole detection operations. The modifications may be found in the revised manuscript at lines 455-457. |
||
|
Comments 5: No relevant experiments to discuss the running time. |
||
|
Response 5: We apologize for overlooking the measurement of the algorithm's runtime. Runtime of the proposed method for measuring a single hole was repeated tested as 2.73s on specific computational platform. The modifications may be found in the revised manuscript at lines 393. |
||
|
4. Response to Comments on the Quality of English Language |
||
|
Point 1: N/A |
||
|
Response 1: |
||
|
5. Additional clarifications |
||

Reviewer 3 Report
Comments and Suggestions for Authors
The paper proposes a method for visually measuring deep holes using a video camera. For image processing, a standard approach in the field of computer vision is used - the method of convolutional neural networks. This method is used in problems of classification, recognition, and image segmentation. The article uses the standard YOLO (You Only Look Once) neural network to recognize multiple objects in an image, as well as the classic Hough transform algorithm associated with identifying objects belonging to a certain class of figures.
It is shown that the image processing algorithm used makes it possible to obtain acceptable accuracy when measuring hole parameters.
However, there are a number of questions about the measurement process and its results.
For a better understanding of the results of the work, it is desirable to add some data and comments.
1. What determines the limitation on the holes being measured (diameter 10 mm and depth 50 mm)?
2. What is the algorithm for aligning the centers of the inlet and outlet holes?
3. Is it possible to measure glare metal holes using this method?
4. Is it possible to measure holes on convex, concave and inclined planes using this method?
5. How does the camera automatically focus on the entrance hole (zoom or camera movement)?
6. How is the tilt (not parallel to the plane of the hole) of the camera compensated?
7. What is the reason for the error in visual measurement of the radius, compared to the coordinatemeasuring machine?
8. How long does it take to characterize one hole?
After adding some data and comments, the article can be accepted for publication.
Comments on the Quality of English LanguageMinor editing of English language required.
Author Response
For research article
|
Response to Reviewer 3 Comments
|
||
|
1. Summary |
|
|
|
Thank you very much for taking the time to review this manuscript. Please find the detailed responses below and the corresponding revisions/corrections highlighted/in track changes in the re-submitted files.
|
||
|
2. Questions for General Evaluation |
Reviewer’s Evaluation |
Response and Revisions |
|
Does the introduction provide sufficient background and include all relevant references? |
Yes |
|
|
Are all the cited references relevant to the research? |
Yes |
|
|
Is the research design appropriate? |
Can be improved |
The experimental methods have been refined, employing more objective approaches to analyze and validate the proposed method. |
|
Are the methods adequately described? |
Can be improved |
The research methodologies employed have been described in greater detail, with the inclusion of new flowchart to illustrate the entire process. |
|
Are the results clearly presented? |
Can be improved |
The experimental results have been re-written, incorporating a larger dataset, with more sophisticated method reproduced and compared. A more impartial algorithm characterization methodology has been adopted. |
|
Are the conclusions supported by the results? |
Yes |
|
|
3. Point-by-point response to Comments and Suggestions for Authors |
||
|
Comments 1: What determines the limitation on the holes being measured (diameter 10 mm and depth 50 mm)? |
||
|
Response 1: Thanks for the question. The demand for visual measurement of deep holes on composite material workpiece addressed in this study arises from the research on robotic automated assembly techniques for composite skin panels of spacecraft sections. This process entails the robot drilling into the interior of the compartments, precisely measuring the positions of assembly holes, and assessing the dimensions of these holes, thereby enabling the inspection of manufacturing quality. In practical assembly of spacecraft, the assembly holes drilled between composite skins and skeletons often have depths of 50mm, diameters of 10mm, and can feature even larger length-to-diameter ratios. |
||
|
Comments 2: What is the algorithm for aligning the centers of the inlet and outlet holes? |
||
|
Response 2: Thanks for the question. The method of aligning the hole centers around the image center is accomplished by employing a region-of-interest (ROI) extraction method, utilizing a pretrained YOLOv3 model. |
||
|
Comments 3: Is it possible to measure glare metal holes using this method? |
||
|
Response 3: Thank you for your valuable suggestion. The proposed hole measurement method primarily consists of the radial penalty Laplacian (RPL) image enhancement algorithm and the inflection point elimination approach. In principle, this method is well-suited for removing the fine-grained noise introduced by surface patterns and the smoothing noise caused by halo effects around the hole outlet. However, the glare around the metal holes is influenced by lighting conditions and the roughness of the workpiece surface, typically exhibiting specific patterns. The proposed method may not optimally suppress the glare for metal holes. Nonetheless, the authors contend that tailored noise reduction algorithms can be designed based on the actual characteristics of the glare. |
||
|
Comments 4: Is it possible to measure holes on convex, concave and inclined planes using this method? |
||
|
Response 4: Thanks for the question. In the visual measurement of holes on curved or inclined surfaces, the contour of the hole typically appears as an ellipse. The proposed method, having employed only circular fitting techniques to extract hole edges, is consequently incapable of addressing such scenarios. This represents a notable limitation of the present study, which has been discussed in the revised manuscript's conclusion section, located at line 455. Nonetheless, the authors posit that the issue could be effectively mitigated by enhancing the current algorithm with more advanced ellipse fitting algorithms, thereby outlining a direction for future research endeavor. |
||
|
Comments 5: How does the camera automatically focus on the entrance hole (zoom or camera movement)? |
||
|
Response 5: We apologize for the confusion caused to the reviewer. In the experimental process of this study, camera focusing was accomplished by manually adjusting the lens. It is important to explain that, in practical in-situ robotic visual measurement applications, we typically employ a technical scheme where a laser displacement sensor is mounted alongside the camera at the robot's end effector. This laser sensor guides the robot's movement, ensuring that the relative distance between the camera and the measured surface remains constant. Such an approach not only eliminates the need to adjust the camera lens's focus but also satisfies the requirements for solving the three-dimensional coordinates of hole positions and diameters. |
||
|
Comments 6: How is the tilt (not parallel to the plane of the hole) of the camera compensated? |
||
|
Response 6: Thank you for raising the question, The reviewer's confusion stems from the authors not having adequately explained the setup process of the experimental platform. The experimental platform incorporates a camera holder that enables independent adjustment of the camera’s pitch, roll and height via its various joints. During the construction of the experimental platform, the camera is initially mounted at an approximate position on the holder, with the workpiece subsequently placed on the platform. Then, the focus of the camera lens and the position of the workpiece are fine-tuned to ensure the camera accurately focuses on the hole of the workpiece surface. At this point, images of the hole are captured using the camera, and the image scale grids are used to visually evaluate if the outline of the tested hole conforms to a perfect circle. Manual micro-adjustments to the camera's pitch and roll are then performed until the tested hole displays as a perfect circle. Thereafter, all joints of the camera holder are tightened to complete the camera installation. The camera is situated roughly 100mm from the workpiece surface, with its optical axis accurately parallel to the target hole axis. The modifications may be found in the revised manuscript at lines 343-354. |
||
|
Comments 7: What is the reason for the error in visual measurement of the radius, compared to the coordinate measuring machine? |
||
|
Response 7: The comparison experimental results validated by Coordinate Measuring Machine (CMM) show that the visual measurement accuracy of the hole size achieves 0.03 mm. The sources of this error can be attributed to two main points: (1) The Basler acA2440-20gm vision system employed in this experiment features a resolution of 2448×2048, providing a field of view approximately 48mm×40mm on the upper surface of the workpiece. This translates to a pixel spacing of roughly 0.02mm. During the measurement process, the circle fitting algorithm proposed in this study may incur a principal error of approximately 1 pixel, which contributes to fitting errors in both hole positioning and diameter measurement. This pixel-level fitting discrepancy is likely a primary source of error in the diameter measurement of the holes. (2) Although the camera's optical axis is nearly parallel to the axis of the measured hole, the lack of perfect alignment results in slight deformation of the imaged hole, which in turn introduces errors during the circle fitting and the three-dimensional calculation processes. |
||
|
Comments 8: How long does it take to characterize one hole? |
||
|
Response 8: We apologize for overlooking the measurement of the algorithm's runtime. Runtime of the proposed method for measuring a single hole was repeated tested as 2.73s on specific computational platform. The modifications may be found in the revised manuscript at lines 393. |
||
|
4. Response to Comments on the Quality of English Language |
||
|
Point 1: Minor editing of English language required. |
||
|
Response 1: Thanks for the suggestion. Authors fully comprehend the importance of English proficiency in meeting the publication standards of the journal, and have thoroughly examined the entire document to guarantee grammatical accuracy and term exactness. Before resubmitting the manuscript, the authors also intend to enlist the services of MDPI's language polishing to ensure its linguistic quality. |
||
|
5. Additional clarifications |
||

Round 2
Reviewer 1 Report
Comments and Suggestions for Authors
Thanks for the responses.
However, some minor corrections are remained:
1. When you have modifications, please precisely show them even in the reference list.
2. In terms of #7 and your response ‘Selection of parameters for the proposed algorithms primarily relies on the characteristics of the images and the underlying principles of the algorithms themselves, hence no extensive optimization measures for algorithm parameters have been undertaken.’, this is somewhat which I was looking for it. This was the main reason why I left the comments. Accordingly, you SHOULD search for suggested keywords and find some distinguished works and guide the readers on the possible advantage of your work over them.
3. In terms of pixel resizing, it obviously influences the image quality. This is a pretty known concept.
4. You also maybe benefit from https://www.sciencedirect.com/science/article/pii/S0898122115005362
5. Concerning the sensitivity analysis, the discussion is deep and don’t want to elongate the process. However, when we discuss sensitivity, it is a ‘what-if analysis’ and it doesn’t matter which method you are using. However, new approaches using intelligent modeling also conceptually are advanced mathematics/statistics. This doesn’t conflict with your applied method.
Good Luck
Comments on the Quality of English LanguagePlease recheck the English
Author Response
For research article
|
Response to Reviewer I Comments
|
||
|
1. Summary |
|
|
|
Thank you very much for taking the time to review this manuscript. Please find the detailed responses below and the corresponding revisions/corrections highlighted/in track changes in the re-submitted files.
|
||
|
2. Questions for General Evaluation |
Reviewer’s Evaluation |
Response and Revisions |
|
Does the introduction provide sufficient background and include all relevant references? |
Can be improved |
|
|
Is the research design appropriate? |
Can be improved |
|
|
Are the methods adequately described? |
Can be improved |
|
|
Are the results clearly presented? |
Must be improved |
|
|
Are the conclusions supported by the results? |
Can be improved |
|
|
3. Point-by-point response to Comments and Suggestions for Authors |
||
|
Comments 1: When you have modifications, please precisely show them even in the reference list. |
||
|
Response 1: Thank you for pointing this out. In accordance with your suggestion, the author has rechecked the manuscript and ensured that all modifications are now indicated using highlighting, including those in the reference list. |
||
|
Comments 2: In terms of #7 and your response ‘Selection of parameters for the proposed algorithms primarily relies on the characteristics of the images and the underlying principles of the algorithms themselves, hence no extensive optimization measures for algorithm parameters have been undertaken.’, this is somewhat which I was looking for it. This was the main reason why I left the comments. Accordingly, you SHOULD search for suggested keywords and find some distinguished works and guide the readers on the possible advantage of your work over them. |
||
|
Response 2: Thank you for raising this question and for your further elaboration. In accordance with the keywords in your suggestion, the authors have conducted an exhaustive search, leading to an analysis of several advanced image denoising algorithms, including the Block-matching and 3D Filtering denoising algorithm (BM3D), Combined Image Denoising in Spatial and Frequency Domains (CID), and the Block-Matching Convolutional Neural Network (BMCNN). The aforementioned advanced denoising approaches, by considering spatial and frequency domain similarity across the entire image, demonstrate remarkable efficacy in suppressing image noise caused by factors such as lens blur or motion distortion. However, it is not particularly advantageous in removing fine and continuous noise caused by the composite material patterns or the smoothing edge interference induced by hole outlet halo. There is also a lack of accurately labeled image datasets for deep holes on composite materials, which are crucial for the pre-training of deep learning models. The authors have revised the manuscript with citations more relevant literature (as shown in reference list [33-35]), and provided an analysis of both the innovations and limitations of the aforementioned algorithms. The modifications may be found in the revised manuscript at lines 171-195. |
||
|
Comments 3: In terms of pixel resizing, it obviously influences the image quality. This is a pretty known concept. |
||
|
Response 3: The authors apologize for having not adequately explained the ROI extraction process to the reviewers within the manuscript, and hereby reiterate the transformation in image size and quality during the ROI extraction process, as illustrated in the following figure. Within this process, downsampling and resizing operations are employed to ensure that the image input into YOLOv3 conform to a uniform size of 416 × 416 pixels. The resultant bounding box defining the ROI from YOLOv3 is described by parameters comprising the center coordinates (cx, cy) and the dimensions (w, h) of the box, all of which are provided in a normalized format. Furthermore, the original image is cropped according to the bounding box, specifically calculated as (2448 * w) pixels in width and (2448 * h) pixels in height, where w and h are the normalized width and height of the ROI from the YOLOv3 output. The operation is executed directly on the high-resolution original image, thereby preserving its resolution intact. The modifications may be found in the revised manuscript at lines 290-292.
ROI (Region of Interest) extraction process
|
||
|
Comments 4: You also maybe benefit from https://www.sciencedirect.com/science/article/pii/S0898122115005362 |
||
|
Response 4: Thanks for the reviewer’s suggestion, and the authors have carefully read the recommended paper. According to the NS-CGNR algorithm proposed in this study, it treats an overall noise suppression in image as the optimal solution to a linear equation pertaining to the original image, and integrates the Tikhonov regularization mechanism to enhance the tolerance for ill-posed problems. The conceptual representation of image processing and the iterative optimization strategy for algorithm parameters of this study have deeply impressed the authors. However, based on the fundamental principles of the algorithm and the denoising efficacy demonstrated in the paper, the proposed method seems to be more suitable for general noise suppression in images caused by factors such as lens blur or motion blur. Conversely, it appears less effective for suppressing the fine noise of the composite material surface in our research, which is marked by non-static positions and significant randomness. Consequently, the NS-CGNR algorithms have not been integrated into this study at this time. |
||
|
Comments 5: Concerning the sensitivity analysis, the discussion is deep and don’t want to elongate the process. However, when we discuss sensitivity, it is a ‘what-if analysis’ and it doesn’t matter which method you are using. However, new approaches using intelligent modeling also conceptually are advanced mathematics/statistics. This doesn’t conflict with your applied method. |
||
|
Response 5: Thanks for your insightful commentary on the sensitivity analysis, the authors fully concur with your viewpoint. While our proposed method centers on traditional parameter tuning approaches, whereas intelligent algorithms provide an alternative lens that can facilitate a deeper comprehension of the model's behavior and uncertainty issues. The authors are dedicated to pursuing further research that explores the potential synergy between these two approaches. |
||
|
4. Response to Comments on the Quality of English Language |
||
|
Point 1: Please recheck the English. |
||
|
Response 1: Thanks for the suggestion. Authors fully comprehend the importance of language quality in meeting the publication standards of the journal. And, with the assistance of colleagues possessing higher level of language proficiency, the entire document has been meticulously reviewed. |
||

Reviewer 3 Report
Comments and Suggestions for Authors
The authors answered all questions and sufficiently revised the article according to the comments.
I believe that the article can be accepted for publication in present form
Author Response
Thank you for the acknowledgment of the manuscript, and the significant efforts made in assisting the authors to enhance the manuscript's quality.